

# Detecting and distinguishing between apicultural plants using UAV multispectral imaging

Alexandros Papachristoforou[1,2,*], Maria Prodromou[3,4,*], Diofantos Hadjimitsis[3,4] and Michalakis Christoforou[4,5]

[1] Department of Food Science and Technology, School of Agriculture, Aristotle University of Thessaloniki, Thessaloniki, Greece
[2] Department of Food Science and Nutrition, School of the Environment, University of the Aegean, Myrina, Greece
[3] Department of Civil Engineering and Geomatics, Cyprus University of Technology, Limassol, Cyprus
[4] Department of Environment and Climate, Eratosthenes Center of Excelence, Limassol, Cyprus
[5] Department of Agricultural Science, Biotechnology and Food Science, Cyprus University of Technology, Limassol, Cyprus
[*] These authors contributed equally to this work.

Corresponding author
Michalakis Christoforou,
m.christoforou@cut.ac.cy

## ABSTRACT

Detecting and distinguishing apicultural plants are important elements of the evaluation and quantification of potential honey production worldwide. Today, remote sensing can provide accurate plant distribution maps using rapid and efficient techniques. In the present study, a five-band multispectral unmanned aerial vehicle (UAV) was used in an established beekeeping area on Lemnos Island, Greece, for the collection of high-resolution images from three areas where *Thymus capitatus* and *Sarcopoterium spinosum* are present. Orthophotos of UAV bands for each area were used in combination with vegetation indices in the Google Earth Engine (GEE) platform, to classify the area occupied by the two plant species. From the five classifiers (Random Forest, RF; Gradient Tree Boost, GTB; Classification and Regression Trees, CART; Mahalanobis Minimum Distance, MMD; Support Vector Machine, SVM) in GEE, the RF gave the highest overall accuracy with a Kappa coefficient reaching 93.6%, 98.3%, 94.7%, and coefficient of 0.90, 0.97, 0.92 respectively for each case study. The training method used in the present study detected and distinguish the two plants with great accuracy and results were confirmed using 70% of the total score to train the GEE and 30% to assess the method's accuracy. Based on this study, identification and mapping of *Thymus capitatus* areas is possible and could help in the promotion and protection of this valuable species which, on many Greek Islands, is the sole foraging plant of honeybees.

# INTRODUCTION

In recent decades, large-scale plant and species identification looked to aerial multispectral sensors because of their ability to provide high-resolution spectral and spatial data (*Lewis*

& Brown, 2001; Paz-Kagan et al., 2019; Robinson et al., 2016). Due to human impacts and climate change, cultivated plants and native vegetation are stressed by various specific factors such as the increase in temperature, extended droughts, salinity, overgrazing and fires (Van den Elsen et al., 2020). This leads to negative effects on food production, the environmental equilibrium and biodiversity. Apiculture with its associated pollination benefits is one of the most important agricultural activities that can affect vegetation, crops production and the environment equilibrium. The importance of the role of bees is being increasingly acknowledged as bees pollinate a significant percentage of the world's crops (Breeze et al., 2011; Khalifa et al., 2021).

Remote sensing systems have become powerful, supporting novel applications in agriculture, but applications in apiculture are, as yet, limited and focused mainly on tracking honeybee predators (Reynaud & Guérin-Lassous, 2016) or honeybee congregation areas (Carl et al., 2017). However, remote sensing could become a useful tool for beekeepers by providing vegetation maps indicating the presence and flowering periods of plants that are highly attractive to bees (Marnasidis et al., 2021) as well as indicating the presence of conventional crops (Donkersley et al., 2014). It can alert beekeepers to the possibilities of transferring colonies to better, safer and more productive locations. As bees' nutrition requirements are very specific (Louargant et al., 2017), detecting and distinguishing apicultural plants from non-apicultural plants and crops is of great importance to beekeepers.

In recent years, classification models used in combination with multispectral and/or hyperspectral aerial images have been deployed to discriminate soil and vegetation types and within vegetation types to distinguish monocotyledonous/dicotyledonous plants (Papp et al., 2021) as well as weeds from crops (Barrero & Perdomo, 2018; Senthilnath et al., 2017). Classification algorithms and models can separate materials and plants using low-altitude, high-resolution spatial images (Rasmussen et al., 2013). Low-altitude images can be collected using aerial platforms such as unmanned aerial vehicles (UAVs) equipped with high-resolution sensors able to detect small objects such as Mediterranean thorny shrubs (Henkin, Rosenzweig & Yaniv, 2014) and measure crop growth by capturing a range of spectral information (Duan et al., 2021). The popularity of UAV technologies has increased rapidly in remote sensing recently because of (i) the ability to collect frequently very precise high-resolution spatial images (Lu & He, 2017; Rango et al., 2006), (ii) the flexibility in flight planning and the rapid acquisition of the images (Torresan et al., 2017) and (iii) the low cost of the equipment (Anderson & Gaston, 2013).

Detecting apicultural plants and distinguishing them from non-apicultural plants is essential for beekeepers, especially in the case of thorny shrubs with similar phenological cycles and close morphological characteristics such as Thymus and Sarcopoterium species. Thymus capitatus (or Corydothymus capitatus) is a small, compact perennial shrub growing to a height of 40–60 cm. It is recognizable for its shape and the long violet flowers held in cone-shaped clusters at the end of the stems. Although in many areas it is considered an evergreen plant, in the Greek Aegean Islands, the abscission of all leaves occurs in autumn and results in a "dry" appearance of the shrub. T. capitatus is considered the most valuable apicultural plant of the Aegean Islands. In many islands (especially the Cyclades Islands

complex) it is the only forage for honeybees; maintenance depends exclusively on the blossom and nectar flow of this species. Furthermore, the superb taste and aroma of thyme honey is highly appreciated by consumers, making it the most expensive Greek honey.

*Sarcopoterium spinosum* (L.), is a spiny, evergreen rosaceous dwarf shrub, 30–60 cm in height, with branches ending in dichotomous and leafless thorns. A combination of the plant's clonal and sexual reproduction contributes to its long-term survival and dominance (*Hill et al., 1998*). Because of its reproductive capability, *S. spinosum* is considered a dominant species in the Mediterranean basin, occupying up to 96% of the total vegetation area in some regions (*Arianoutsou & Margaris, 1982*). The *S. spinosum* can resprout and grow faster than the apicultural plants after destruction, *e.g.*, by fire (*Niphadkar & Nagendra, 2016*). It may then colonize land traditionally covered by *Thymus* spp. and thereby have a negative impact on apiculture, especially for beekeepers who produce and sell value-added honey labeled as "thymus honey".

Systematic monitoring could be of great importance for apiculture management to enable beekeepers to detect and estimate the areas covered by *Thymus* species, to distinguish thyme and other apicultural plants from non-apicultural plants and to take measures to respond to changes. It is also mandatory to keep track of newer invasions (*Hestir et al., 2008*) of *S. spinosum* within wild *T. capitatus* ecosystems and other apicultural plant species, especially after a wildfire.

Various methods are described in the literature for the classification of ground "features" such as plants (weeds, bryophytes and invasive plants) (*Ishida et al., 2018*; *Papp et al., 2021*; *Rasmussen et al., 2013*). The methods referred in the literature for detection and identification of apicultural plants (or invasive species/weeds) vary. Some include the Random Forest (RF) (*Dmitriev et al., 2022*; *Große-Stoltenberg et al., 2016*; *Hill et al., 2017*; *Kattenborn et al., 2019*; *Sheffield et al., 2022*; *Singh & Singh, 2022*), Classification and Regression Trees (CART) (*Fariz et al., 2022*; *Ishak et al., 2008*; *Raj & Sharma, 2022*; *Traganos et al., 2018*), Support Vector Machine (SVM) (*Forster et al., 2017*; *Paz-Kagan et al., 2019*; *Skowronek, Asner & Feilhauer, 2017*), Mahalanobis Minimum Distance (MMD) (*Sampedro & Mena, 2018*, *Yang & Everitt, 2010*) and Gradient Tree Boost (*Sujud et al., 2021*). For comparative purposes, the majority of the studies have used multiple supervised classification models (*Arasumani et al., 2021*; *Gašparovičová, Ševčík & David, 2022*; *Zhu et al., 2022*) such as Support Vector Machines (SVMs), Artificial Neural Network (ANN), Gradient Tree Boost (GTB) and Random Forest (RF) classifiers. They have been widely applied for the detection and identification of plants, in combination with the use of UAV high-resolution aerial images with (*Barrero & Perdomo, 2018*; *Bolch, Hestir & Khanna, 2021*; *Pretorius & Pretorius, 2015*). The accuracy of the data retrieved from individual pixels (the spatial units in an image) are analyzed by the spectral information they contain and compared with ground-truth information (*Google, 2022*).

Through remote sensing techniques, the present study aims to detect, identify and distinguish *T. capitatus* from the non-apicultural, fast-growing and aggressive plant *S. spinosum*. Remote sensing will be used for preliminary mapping and classification of apicultural areas. We conducted the present study on Lemnos (Greece), an island of great apicultural interest where expansion of *S. spinosa* is increasing rapidly and replaces

the apicultural vegetation, especially after wildfires which pose a serious threat for the beekeeping community of the island. A UAV equipped with a high-resolution multispectral camera was used, and pixel-based supervised classification algorithms (Gratiend Tree Boost, CART, Random Forest, Mahalanobis Minimum Distance and Support Vector Machine) were implemented on the UAV orthophotos and processed in the GEE platform (*Google, 2022*) along with ground-truth information. The GEE platform was used for image processing since it is low-cost, accessible and user-friendly. The comprehensive review by *Yang et al. (2022)* proves the incorporation of artificial intelligence (AI) models into to GEE with the most common AI models used for vegetation and/or crop mapping are the RF, CART and SVM. For the evaluation of the AI models the Producer's Accuracy (PA), User's Accuracy (UA), Overall Accuracy (OA) and Kappa coefficient are mostly used.

Through the GEE it is possible to implement further supervised classification methods that also provide deep learning methods, such as fully convolutional neural networks (FCNN), or deep neural networks (DNN), but these algorithms require external training on the TensorFlow platform (*Abadi et al., 2016*). Our study focuses entirely on the use of open-access solutions and for this, the traditional machine-learning algorithms were selected because those of Google Cloud are billable and require much more training data than machine learning (*Yang et al., 2018*).

## MATERIALS & METHODS

### Study area

The research was conducted in three experimental sites on Lemnos Island (Fig. 1) which lies in the northern part of the Aegean Sea in Greece (SA1: 25°19′10″E, 39°48′43″N, SA2: 25°18′39″E, 39°48′24″N, SA3: 25°19′5″E, 39°48′50″N). The historical fire events of the same plots were identified through Fire Information for Resource Management System (*NASA, 2021*) and downloaded as active fire and thermal anomaly locations using MODIS and Visible Infrared Imaging Radiometer Suite (VIIRS) platforms to ensure that no fire was misreported at the specific plots.

The aerial and ground-truth measurements were collected on the 21st of October 2020. During the autumn, *T. capitatus* is dry and can be recognized and distinguished from *S. spinosum* through visual observation.

The vegetation in Lemnos is mainly low, with shrubs, bushes, and flowers, and includes the Aegean Sea plants such as thyme, oregano, and other herbs (*Panitsa et al., 2003*). Because it is a volcanic island, there is no forest but there are extensive fertile plains. Lemnos is characterized by a Mediterranean climate with mild winters, and a significant characteristic are the strong winds, especially during August and winter.

### UAV Images acquisition and processing

A DJI P4 multispectral drone was used for the missions on Lemnos Island. The P4 multispectral is a tetra-copter rotary-wing aircraft with vertical takeoff and landing and can hover at low altitudes. It is equipped with one RGB sensor for visible light imaging and five monochrome sensors—BLUE (450 nm ± 16 nm), GREEN (560 nm ± 16 nm), RED (650 nm ± 16nm), RED EDGE (730 nm ± 16 nm) and NIR (840 nm ± 26 nm)—for

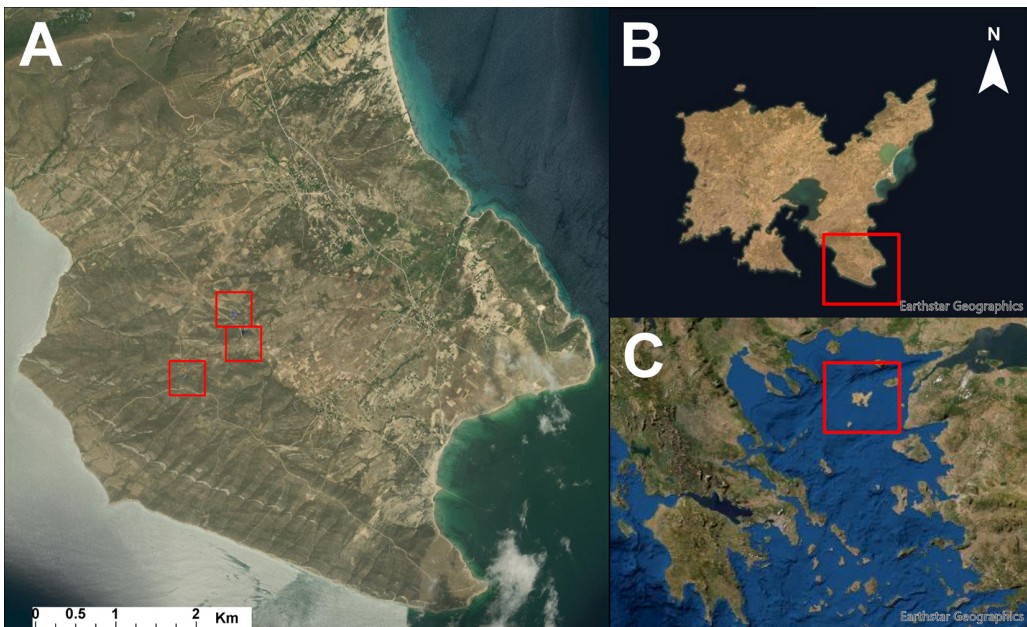

**Figure 1** **Location of the sample plots.** (A) Sample plots in southeast part of the island. (B) Lemnos Island and the experimental area. (C) the island of Lemnos in the northeast Aegean Sea, Greece. Source: Esri, Maxar, Earthstar Geographics, and the GIS User Community.

multispectral imaging plus a spectral sunlight sensor on top of the aircraft to compensate images while detecting the solar irradiance in real time. Flight paths were designed using DJI GSpro software, and missions were conducted between 11:00 am and 14:00 pm to reduce shading and at 20 m altitude, 7–8 m/s wind speed, 90° course angle, 70% front overlap ratio and 60% side overlap ratio. The P4 multispectral camera was set to capture images at regular intervals parallel to the aircraft's flightline. Three orthophotos were derived from 3,138 images acquired using a multispectral UAV with a ground sampling resolution of 0.8 cm for the classification method.

### Ground reference survey

For this study, the training sample plots were chosen from three different regions on Lemnos Island using the DJI P4 multispectral UAV and orthophotos created in DJI TERRA software and analyzed manually in ArcGIS Pro software. The samples were classified into three main types: Thyme, Sarcopoterium, and Others. Within each sample plot, a sub-plot was visually identified and used as a training data set for the three selected classes. Ground points were added using a GNSS sensor and used as known targets. Orthophotos and ground point targets were added in the ArcGIS Pro software for the creation of shapefile polygons required by the GEE.

### Data processing

For this study the GEE was used for the UAV image processing. The GEE is a planetary-scale platform for scientific analysis and visualization of geospatial datasets. In this platform,

| Table 1 Kappa value interpretation, (*Cohen, 1960*). | |
|---|---|
| **Kappa** | **Interpretation** |
| 0 | No agreement |
| 0,0–0,20 | Slight agreement |
| 0,21–0,40 | Fair agreement |
| 0,41–0,60 | Moderate agreement |
| 0,61–0,80 | Substantial agreement |
| 0,81–0.99 | Near perfect agreement |
| 1 | Perfect agreement |

the open-source images acquired by several satellites are accessible and can be efficiently imported and processed in the cloud without the necessity of downloading the data to local computers. Several image-driven products and many remote sensing algorithms, including classification algorithms and cloud masking methods, are also available on this platform (*Gorelick et al., 2017*; *Kumar & Mutanga, 2018*).

The first stage focuses on the processing of drone images to create GeoTIFFs, the format required by the GEE, for each case study. The second stage is the collection of the training sample for each category. The third stage includes the image classification, and the final steps focus on the products' accuracy assessment.

## Vegetation indices

In this study, we tested how different drone band combinations, referring both to single spectral bands and spectral indices can affect the classification results and which can be considered as the best combination resulting in higher thematic accuracy. According to the literature, existing studies have shown that the use of spectral indices can increase classification accuracy (*Phan, Kuch & Lehnert, 2020*; *Praticò et al., 2021*; *Yang et al., 2021*). The spectral response of apicultural plants was estimated using six commonly used vegetation indices (Vis). They were calculated from multispectral images using the equations presented in Table 1. The spectral indices used in this study were added as a new layer (band) to the image derived from the UAV.

## Supervised classification of UAV orthophotos

For the implementation of the pixel-based supervised classifications, training samples were collected for the three study areas. The samples were allocated to one of three categories: Thyme, Sarcopoterium and Other. In the Other category, we included features such as bare soil, shadows, dry and other vegetation types. Samples were imported into the GEE platform to undergo supervised classification of the orthophotos for each study area, SA1 (Fig. 2A), SA2 (Fig. 2B and SA3) (Fig. 2C).

The GEE platform uses a classifier package that provides supervised classification by several different traditional machine learning (ML) algorithms, running in Earth Engine. In this study, we compared the performance of Random Forest (RF), Classification of Regression Trees (CART), Gradient Tree Boostbased (GTB), Mahalanobis minimum-distance (MMD), and Support Vector Machine (SVM) for their reliability in species classification.

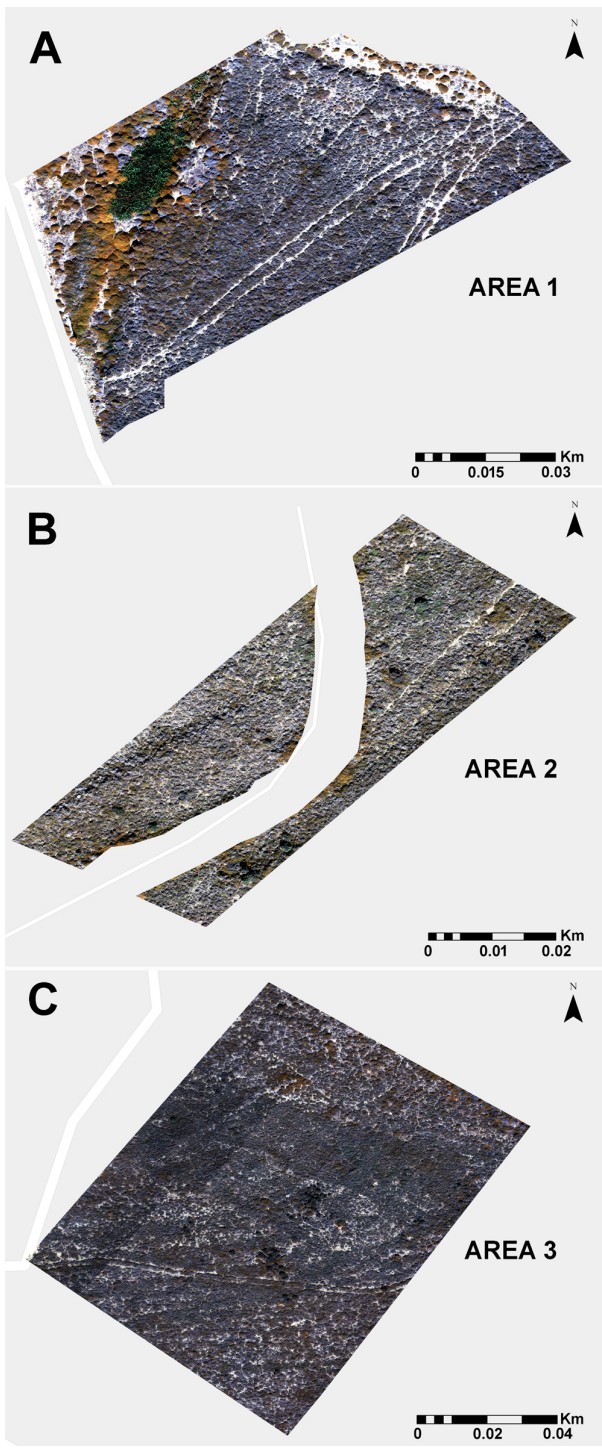

**Figure 2** **The three sample plots used for the detection and dinstiction of *Thymus capitatus* and *Sarcopoterium spinosum* species.** (A) Study area 1. (B) Study area 2. (C) Study area 3.

CART is a non-parametric tree-based machine learning predictive algorithm used for training data to generate prediction models. CART algorithm is based on a decision tree classifier, which is made by a set of training data. CART, developed by *Breiman (1984)*, facilitates simple decision making and regression analyses, and is used for training data to generate prediction models. CART algorithm runs by separating nodes until it reaches the terminal nodes. CART is based on a decision tree where the producer models are acquired by recursively partitioning the data space and fitting the classification model in each part to predict its class. CART also uses the surrogate separator to make the best use of data with missing values. The advantage of CART is its ability to deal with a variety of types of data, producing a complex decision flowchart like a tree structure through learning the input dataset (*Breiman et al., 2017*; *Nakhaei, Tafreshi & Saadi, 2023*). CART, as previously described in *Neetu & Ray (2019)* is "an advanced technique based on Decision Tree (DT) classifier, which is built from a set of training data. The advantage of CART is that it is simple in understanding, visualizing and interpreting. Both numerical and categorical data can handle in CART".

RF is a tree-predictor algorithm whereby each object depends on the values of a random vector sampled independently and with the same distribution for all objects (*Breiman, 2001*). The RF has the advantage of combining multiple decision trees and using average or voting schemes to calculate the final results. This contrasts with a single decision tree and helps to reduce the instability and sensitivity of the model while increasing its predictive power (*Vorpahl et al., 2012*). Using the tree rule, the decision-making process is repeated at each internal node from the root node until the given termination condition is satisfied (*Ließ, Glaser & Huwe, 2012*).

GTB is a mass tree-based algorithm (*Friedman, 2002*) that classifies the train weak learners assigning them to stronger weights by adding new objects which decrease prediction errors (*Callens et al., 2020*; *Ustuner & Balik Sanli, 2019*). A set of logical if-then conditions (instead of linear relationships) from the classification method and tree regression are then used to predict and classify the result. The GTB method provides the capabilities of two algorithms: (a) regression tree models that respond to the prediction where they describe the operators through optimal and boosting binary separation, and (b) an adaptive method that achieves the combination of many simple models that helps to obtain higher performance. It is an additive model where the gradient identifies the shortcomings of the previous models. The residuals of the current classifier are used as input to the next one in which trees are built, thus capturing the variation in the data (*Elith, Leathwick & Hastie, 2008*).

The Mahalanobis distance classifier is a non-parametric and distance-based classifier that assumes that all class fluctuations are equal. All pixels are classified in the nearest class (training sample) unless a distance limit is set, in which case some pixels may not be classified if they do not meet the limit (*Hengst, 2020*). Specifically, the Mahalanobis distance classifier has three main steps: (1) computation of the mean vector in each band based on the application of the training sample data; (2) calculation of the spectral distance between the mean vector of each sample and the measurement vector of the candidate

pixel; and (3) the pixel with the smallest spectral distance is classified into the nearest class (*Perumal & Bhaskaran, 2010*).

SVM, proposed by *Cortes & Vapnik (1995)*, is a widely used algorithm for classifying data. It attempts to create hyperplanes that best divide a dataset into different classes. The best hyperplane is one that maximizes the margin between the closest data points from each class, which are called support vectors (*Bishop, 2006*). However, the method only works for datasets that can be separated linearly. To handle non-linear datasets, a non-linear kernel function is used to transform the data into a high-dimensional space where it becomes linearly separable. The most commonly used kernels are polynomial and radial basis function kernels. The formula for linear SVM is an optimization problem that has an objective function which is the minimization of $min_{w,\xi_i,b}\left\{\frac{1}{2}||w||^2 + c\sum_{I=1}^{M}\xi_i\right\}$ with the constraint that

$$y_i((\varphi(x_i), w+b)) \geq 1 - \xi_i, \xi_i \geq 0, i = 1, \ldots, N.$$

The $y_i$ is the class label, the $\xi_i$ presents the positive slack variables, the C is the regularization and b is the bias term (*Oommen et al., 2008*; *Cortes & Vapnik, 1995*).

## Accuracy assessment of image classification and statistical analysis

For the classification, 70% of the total sample was used for training the GEE classifiers while the remaining 30% was used for evaluation purposes (accuracy assessment). The evaluation was achieved based on confusion matrix statistics allowing the calculation of the user's accuracy, producer's accuracy, Kappa value and F-score per classifier.

Confusion matrix is estimated by comparing reference and classified results per thematic category (*i.e.,* class). Producer's accuracy estimates how well the reference pixels per class are correctly classified, while user's accuracy estimates the probability that a pixel classified into a class represents the same (true) class using the reference sample (*Lee et al., 2016*; *Lu & Weng, 2007*; *Praticò et al., 2021*). The Kappa value is a measure of the classification performance compared to values assigned by chance (*Lu & Weng, 2007*). It can take values from 0 to 1, in which a Kappa coefficient close to 0 indicates no agreement between the classified results and the reference (truth) data, while a Kappa coefficient close to 1 indicates an agreement between classified and reference (truth) data. Finally, the so-called F-score, known as the harmonic mean of recall and precision, is another classification accuracy metric with the same meaning as producer's accuracy (PA) and user's accuracy (UA) respectively (*Praticò et al., 2021*).

Kappa Coefficient equation used for the training of the GEE classifiers.

$$OA = \frac{1}{N} \sum_{i=j=1}^{n} C_{ij}$$

$$PA = \frac{C_{ij}}{\sum_{i=j=1}^{n} C_j}$$

$$UA = \frac{C_{ij}}{\sum_{i=j=1}^{n} C_i}$$

$$F = 2 * \frac{PA * UA}{PA + UA}$$

$$Kappa\ Coefficient = \frac{N \sum_{i=j=1}^{n} C_{ij} - \sum_{i=j=1}^{n} C_i C_j}{N^2 - \sum_{i=j=1}^{n} C_i C_j}$$

Where,

N Number of rows in error matrix

$C_{ij}$ Number of observations in row and column

$C_i$ Total number of observations in row

$C_j$ Total number of observations in column

N Total number of observations

PA Producer's accuracy

UA User's Accuracy

## RESULTS

### Accuracy assessment

For detecting and distinguishing apicultural plants such as *T. capitatus* from non-apicultural plants, such as *S. spinosum*, we trained samples of *T. capitatus* and *S. spinosum* using the classification accuracy assessment derived from the confusion matrix. Different band compositions (DJi bands + spectral indices) were implemented for each case study, and the GEE provided the same training samples and validation data for five different classifiers.

The evaluation of the classification results accuracy for each classifier as well as for each band combination with spectral indices, the PA, UA and F-score parameters, were estimated for the three classes separately. The total accuracy estimation was derived from each classifier and each combination based on the OA and Kappa parameters. Figure 3 presents the results for each study area where the stacked columns show the PA, UA and F-score parameters for each class separately. PA, UA and F-score parameter give values from 0 to 1, so if the three parameters together are close to the value three, they have achieved high values in the classification. When the score approaches the value two on the *Y* axis, the classifier achieves the higher accuracy from the OA and Kappa parameters.

For the SA1 study area, the RF and GTB classifiers presented the highest accuracy when using the combinations *ALL BANDS + VEGETATION INDICES, VIS + NIR + RE and VIS + NIR bands,* which presented the same behavior with kappa values = 0.97 and OA = 0.98 in all cases. The lowest accuracy was presented by the SVM classifier when combined with

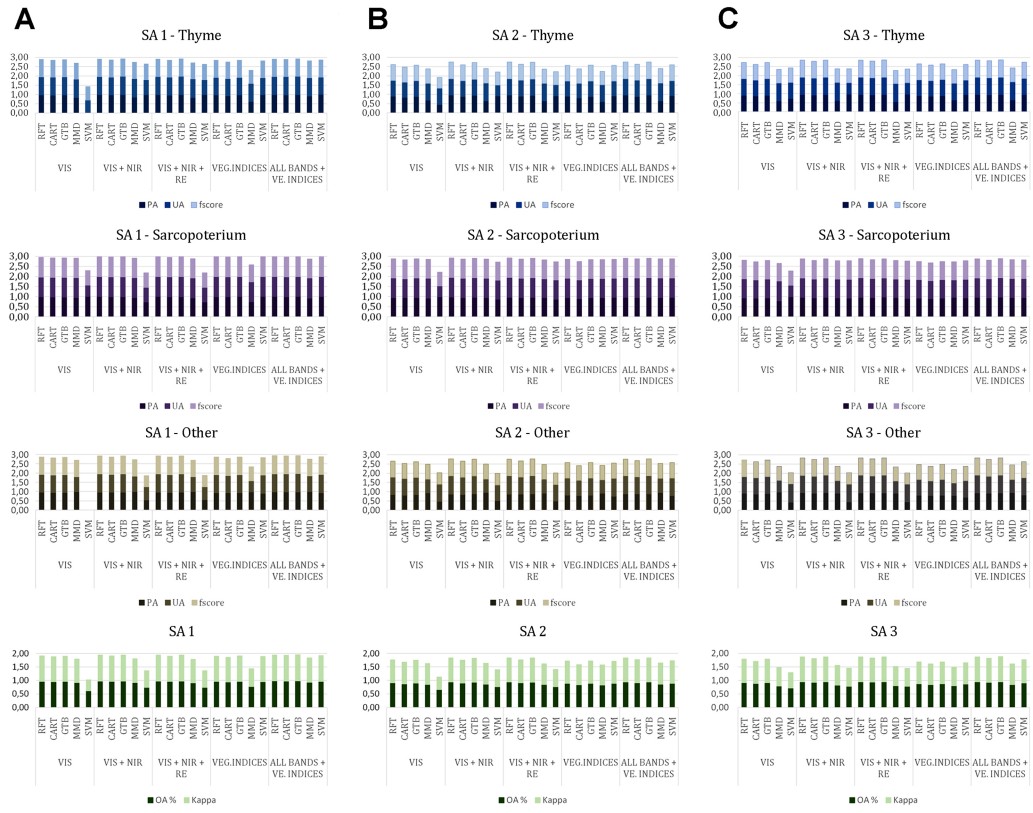

**Figure 3** **Classification performance of the five classifiers.** RFT, Random Forest; CART, Classification and Regression Trees; GTB, Gradient Tree Boost; SVM, Support Vector Machine; MMD, Mahalanobis Minimum Distance) in Google Earth Engine using the Producers Accuracy, Users Accuracy and f-score for each class separately (thyme, blue charts; Sarcopoterium, purple charts; and Other, grey charts) and the Overall accuracy and Kappa coefficient with green charts for the three study sites (SA, Study Area). A, B, C: Stusy Area 1, 2, 3, correspondingly, VIS, Visible spectrum; NIR, Near Infrared; RE, Red Edge; VEG.INDICES, Vegiatation indices.

the VIS classifier (Kappa =0.42 and OA =0.61) and with the *VIS + NIR, VIS + NIR + RE* where in both cases they scored Kappa =0.62 and OA =0.74 values. Low performance was also presented by the MMD classifier with performance Kappa =0.66 and OA =0.77.

For the SA2 study area, the highest accuracy was achieved using the RF classifier based on the combinations *VIS + NIR + RE, VIS + NIR plus ALL BANDS + VEGETATION INDICES*, and GTB with the combination *ALL BANDS + VEGETATION INDICES*, where they presented similar behavior, with Kappa values =0.90 and OA =0.93 in all cases. The lowest accuracy was presented by the SVM classifier mainly with the VIS combination (Kappa =0.47 and OA =0.65). The MMD classifier also showed low performance. In this case, the CART classifier also showed a relatively low performance when the *VEGETATION INDICES* combination was used providing Kappa = 0.75 and OA = 0.83 values.

In the third study area (SA3), the highest accuracy was provided by GTB and RF classifiers when combined with *ALL BANDS + VEGETATION INDICES, VIS + NIR + RE* and *VIS + NIR* with Kappa = 0.92 and OA = 0.95 values. The lowest accuracy was presented

by the SVM classifier when combined with the VIS (Kappa =0.58 and OA =0.72). Low performance was achieved when the SVM classifier was combined with the *VIS + NIR + RE* and *VIS + NIR*. The MMD classifier also performed low accuracy (Kappa 0.68−0.73 and OA =0.79−0.82).

## Classification performance between thyme, Sarcopoterium and other in Google Earth Engine

For the three case studies, the total accuracy for the five classifiers (RF, GTB, CART, MMD and SVM), was calculated using the average value from the Kappa and OA parameters. On average the GTB and RF classifiers presented similar behavior with Kappa =0.91 and OA =0.94 values. Also, the average of the CART classifier presented accuracies that belong to the "near-perfect agreement" category with the average values for Kappa and OA amounting for 0.88 and 0.92, respectively. The classifiers which presented the lowest average were SVM (Kappa = 0.70 and OA =0.80) and MMD (Kappa =0.78 and OA =0.83). In addition, the average of the accuracy (obtained from the various combinations of spectral channels and vegetation indices was calculated with the combination of *ALL BANDS + VEGETATION INDICES*), scored the highest accuracy (Kappa =0.90 and OA =0.93). The combinations VIS + NIR and VIS + NIR + RE gave similar results with an average accuracy of Kappa =0.84 and OA =0.89, while the combinations *VEGETATION INDICES* presented average accuracy of Kappa =0.82 and OA =0.88. The VIS presented average accuracy of Kappa =0.78 and OA =0.85 providing the lowest returns on average (Fig. 3).

To sum up, the findings of the present study show that the GTB and RF classifiers together with the *ALL BANDS + VEGETATION INDICES* combination is considered ideal for the classification of beekeeping plants and more specifically for the separation of *T. capitatus* and *S. spinosum.*

## Detection and distinguishing maps

The classification maps retrieved from GEE after the selection of the GTB classifier for the detection and distinguishing of the two shrub species (Thyme and Sarcopoterium) are presented in Fig. 4. In the three sample plots (SA1, SA2 and SA3), the *T. capitatus* plants are shown in purple, *S. spinosum* plants in yellow, and all the other vegetation species and bare soil are shown in black. The roads are masked out to improve the accuracy of the results. Based on these maps for each case study, the total area covered by each class was estimated. The area covered by each species is shown in Fig. 4. The evaluation of the total area for each class was for SA1: Thyme = 0.21ha, Sarcopoterium = 0.10 ha and Other = 0.07 ha (Fig. 4A), SA2: Thyme = 0.13 ha, Sarcopoterium = 0.04 ha, other = 0.02 (Fig. 4B), SA3: Thyme = 0.52 ha, Sarcopoterium = 0.26 ha and Other = 0.15 ha (Fig. 4C). These figures were derived using the GTB classifier.

While running the Kappa coefficient algorithm, known sample points and combinations of UAV bands and VIs were added to the algorithm until the Kappa coefficient reached 81% or higher. Of the five classifiers used in this study (RF, GTB, CART, MMD and SVM), the algorithms which are based on the tree-based decision give the best outcomes. The

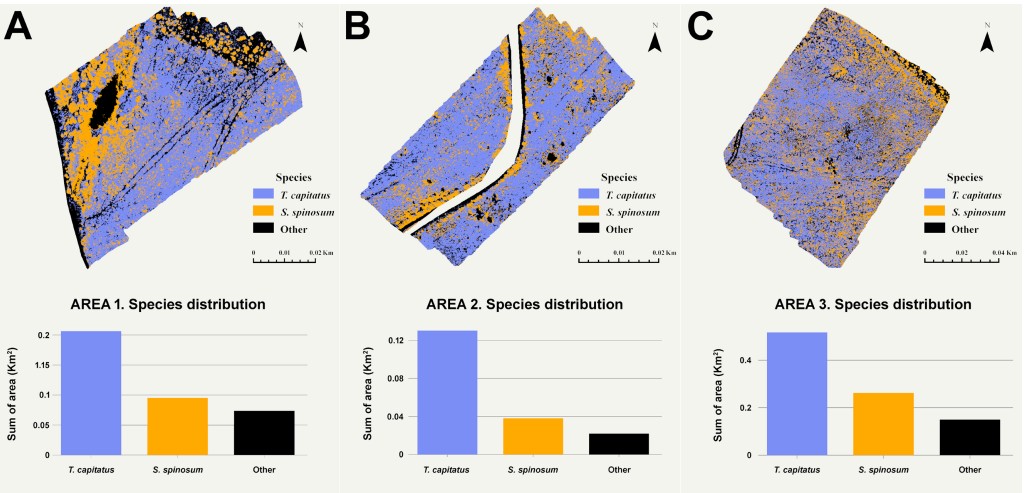

**Figure 4** Classification maps retrieved from GEE after the selection of the GTB classifier for the detection and distinguishing of the two shrub species. Evaluation of area covered by *Thymus capitatus* and *Sarcopoterium spinosum* in (A) study area 1, (B) study area 2 and (C) study area 3.

Gradient Tree Boost and the RF classifier achieved the higher Overall Accuracy and Kappa coefficient value when used for the three sample plots but also when used among all five UAV bands and the five VIs combinations. Based on the results, for the first three classifiers with the best performances, a variable importance chart was created. As shown in Fig. 4, the variable importance shows that RF classifier uses all the variables (UAV bands and VIs), in contrast to GTB and CART classifiers, in which the RVI and CI-RE indices do not contribute to the prediction analysis. The variable importance plot is a very useful tool to present the variables that can make accurate predictions when using the model.

## DISCUSSION

The present study aimed to investigate the feasibility of detecting and distinguishing apicultural plants from non-apicultural plants using UAV aerial multispectral images through GEE platform in traditional beekeeping areas, such as Lemnos Island. Results showed that such detection and identification is possible. From the five classification algorithms (RF, CART, GTB, MMD and SVM) used to train the classification model, RF had the highest accuracy values and this is confirmed by *Pérez-Cutillas et al. (2023)* who found that among the non-parametric classification methods the most frequently used algorithm was Random Forest, with 31% of the cases used. This study employs five different machine-learning algorithms because the existing literature is contradictory in terms of the classification accuracy of various classifiers for vegetation species. For example, *Elkind et al. (2019)* used Random Forest models on the GEE platform to classify buffelgrass in Saguaro National Park with an average overall accuracy of 93%. However, other studies have found different classifiers to have better performance. For example, *Sujud et al. (2021)* found that the SVM classifier had the highest overall accuracy for cannabis detection, while *Mustak et al. (2019)* found that SVM outperformed RF and CART for crop discrimination.

In contrast, *Wahap & Shafri (2020)* found that CART had the best accuracy for land cover monitoring compared to RF and SVM.

The performance of the three classifiers along with the evaluation of the different UAV bands and Vegetation Indices and their combination for the detection and distinguishing of *T. capitatus* and *S. spinosum* were the main goals of this study. The resulting metrics presented high accuracy values for all three classifiers, with RF providing the highest values for overall accuracy and Kappa coefficients. The overall accuracy in distinguishing *T. capitatus* from *S. spinosum* when using RF was OA = 98% and Kappa = 0.97 for SA1, OA = 93% and Kappa = 0.90 for SA2, and OA = 95% and Kappa = 0.92 for SA3. As a result, it was possible to distinguish between the apicultural and non-apicultural species.

In apiculture, UAVs have been used as a tool for monitoring drone congregation areas (DCAs—where male honeybees congregate for mating) (*Carl et al., 2017*) to provide useful information for honeybee queen-rearing and breeding. Furthermore, UAVs have been used for the tracking of hornets *Vespa velutina* (*Reynaud & Guérin-Lassous, 2016*). *V. velutina* is an invasive species in Europe (*Rortais et al., 2010*) and can cause severe losses in *Apis mellifera* colonies (*Arca et al., 2014*). Being able to track such predators could help in the control of their spread and limit their impact on honeybees. The use of UAVs equipped with multispectral sensors is nowadays used as a tool for monitoring invasive plants (*Elkind et al., 2019*) and wild plant species in inaccessible areas and with zero environmental disturbance (*Questad et al., 2022*). Success in identifying Thyme plants, the dominant apicultural plant of Lemnos Island and most Greek Islands, is of great importance for beekeeping in Greece. Even though *T. capitatus* is protected by Greek legislation, many Thyme areas are destroyed every year (by farmers, herdsmen *etc.*) on the assumption that they are not Thyme areas but just areas of scrub vegetation. Fire services usually claim that they cannot recognize the difference between a Thyme plant and other scrubs and, as a result, they cannot take any action against the destruction of T. capitatus areas. Our work could be extended to applications such as the mapping of T. capitatus areas in the entire country and lead to the monitoring and protection of such areas. Furthermore, since competition between apicultural plants and non-apicultural plants is intensified as a result of plant stress caused mainly by climate change and wildfires (*Kosmas et al., 2015*), the identification and protection of Thyme areas will help protect honeybee populations that are under threat. Previous research (*Vieira, Cianciaruso & Almeida-Neto, 2013*), investigated the consequences of plant extinctions on pollinators and produced prediction models in which changes in plant diversity (especially the extinction of *T. capitatus*) could lead to the coextinction of pollinators (including honeybees). The encroachment of Sarcopoterium in areas of Thymus species is not easily detected or managed during the early invasion stages because of the morphological similarity of the two species.

Remote detection and distinguishing of the two species using UAV images and machine learning techniques might allow researchers, beekeepers, and state authorities to monitor the growth and early expansion of *S. spinosum* over *T. capitatus* after a catastrophic fire events and provide real-time data enabling timely control measures. Accurate estimation of *T. capitatus* distribution is fundamental in maintaining and developing the quantity and quality of the Lemnian honey. It would also benefit beekeepers who need to move their

hives in search of *T. capitatus* blossom. In a similar study, UAV aerial images have been used to estimate flower numbers and their potential nectar yield, and to identify areas of *Robinia pseudoacacia* suitable for honeybee foraging (*Carl et al., 2017*).

The continuing monitoring of historic Thymus lands will provide valuable environmental information and enable estimates of Lemnian wild Thyme honey production.

## CONCLUSIONS

The aim of the present study was to detect apicultural plants and to distinguish them from plant species not suitable for apiculture using a pixel-based approach. It explored the capability of the GEE cloud platform for apicultural plant species classification. For this purpose, *T. capitatus* and *S. spinosum* were chosen and scanned using a five-band multispectral drone in three areas of Lemnos Island.

From the five classification algorithms (RF, CART, GTB, MMD and SVM) used to train the classification model, the RF and GTB classifiers in combination with the *ALL BANDS + VEGETATION INDICES, VIS + NIR + RE* and */or VIS + NIR bands,* provided the highest accuracy values as shown in the Kappa coefficient and overall accuracy, in all three study sites and manage to detect and distinguish *T. capitatus* from *S. Spinosum.* As described in the literature, the RF and GTB classifiers were effective for distinguishing the two species and thus should be used to identify extensive areas of apicultural plants and to monitor bee habitats. This research confirmed that cloud computing architectures in combination with machine learning algorithms can very accurately and quickly identify invasive species.

The use of a tetra-copter drone flying at a low altitude and equipped with a multispectral camera can provide high resolution images. However, multiple flights are necessary. To overcome this limitation, we propose the use of a vertical take-off and landing (VTOL) UAV with a high-resolution coverage. According to *Awad (2018)*, the multispectral images retrieved from satellites do not provide complete separation of forest plant species and thus the hyperspectral sensor is proposed as a better plant species separator due to its almost continuous spectra, covering spectral details that might pass unnoticeable when using the multispectral sensor (*Adão et al., 2017*). A limitation of hyperspectral sensors attached on a UAV is their high cost compared to a low-cost multispectral sensor. A tetracopter UAV is low cost and suitable for small areas but costly for extensive mapping because of the need for multiple flights. A further study will use of VTOL UAV for the detection of other apicultural plants and to distinguish them from invasive plant species in the beekeeping areas in Greece and other Mediterranean regions.

## ACKNOWLEDGEMENTS

The authors acknowledge the Agricultural and Apicultural Cooperation of Lemnos Beekeepers and especially their president, Dimitris Paleologos for his assistance during our field work, the 'EXCELSIOR': ERATOSTHENES: EXcellence Research Centre for Earth Surveillance and Space-Based Monitoring of the Environment H2020 Widespread

Teaming project (www.excelsior2020.eu) for providing the equipment for the experiments, and Stephen Fleming for editing the manuscript.

### Funding

The 'EXCELSIOR' project (EXCELSIOR': EXcellence Research Centre for Earth Surveillance and Space-Based Monitoring of the Environment H2020 Widespread Teaming project (www.excelsior2020.eu)) has received funding from the European Union's Horizon 2020 research and innovation programme under Grant Agreement No 857510, from the Government of the Republic of Cyprus through the Directorate General for the European Programmes, Coordination and Development & the Cyprus University of Technology. The funders had no role in study design, data collection and analysis, decision to publish, or preparation of the manuscript.

### Grant Disclosures

The following grant information was disclosed by the authors:
EXCELSIOR.
European Union's Horizon 2020: 857510.
Government of the Republic of Cyprus through the Directorate General.

### Competing Interests

The authors declare there are no competing interests.

### Author Contributions

- Alexandros Papachristoforou conceived and designed the experiments, performed the experiments, analyzed the data, prepared figures and/or tables, and approved the final draft.
- Maria Prodromou conceived and designed the experiments, performed the experiments, prepared figures and/or tables, and approved the final draft.
- Diofantos Hadjimitsis analyzed the data, authored or reviewed drafts of the article, and approved the final draft.
- Michalakis Christoforou conceived and designed the experiments, performed the experiments, analyzed the data, authored or reviewed drafts of the article, and approved the final draft.

### Data Availability

The raw data are available in the Supplemental Files.

### Supplemental Information

Supplemental information for this article can be found online at http://dx.doi.org/10.7717/peerj.15065#supplemental-information.

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
