# Peer review of "Detecting and distinguishing between apicultural plants using UAV multispectral imaging"

_PeerJ, doi:10.7717/peerj.15065_

## Round 0.1 · original submission · Minor Revisions

The reviewers have provided some minor changes in the paper to proceed. Please incorporate all the given comments. Please provide the methodology and clarity of requested changes in methodology and please provide policy implications. Moreover, please improve the language of the paper.

Reviewer 1 ·

Basic reporting

See my comments

Experimental design

See my comments

Validity of the findings

Findings are well presented but not justified. See my comments

Additional comments

Title: I have concern on word "discrimination". It usually use in negative sense. Is this the case in your apiculture?
Abstract: Is there any policy recommendation to relevant stakeholders based on your findings. If so, write a sentence on the Abstract and elaborate and in results and discussion.
Introduction: Introduction is written nicely. Accept my appreciation. However, why this study is necessary in Greece context is needed to address. In short, write the knowledge gap and how you will fill it.
Material and method: It is written nicely. Incorporate the following comment;
You used confusion matrix statistics such as the user's accuracy, kappa value and F-score, per classifier (line 230). Elaborate these statistical methodologies.
Results: Results are well presented. But, these can be improved further. For instance, your discussion is just listing of past literature. Justify your each result from these past literature (discussion). It will give you new insights based on your findings and hence policy implication if any. This can be incorporate here as well as in the Abstract

Reviewer 2 ·

Basic reporting

Abstract section missed the practical implications of the study.
Introduction needs to add the net contribution of this study in the current state of literature. For this purpose, please write a summary in the last paragraph of the introduction section.
The discussions of the results are fine and well-compared.

Experimental design

Methodology is well-written.

Validity of the findings

The validity of the findings should be added by some diagnostic test.

Additional comments

The conclusion section should be added with implication of the study, limitations of the present study and future direction.

Reviewer 3 ·

Basic reporting

The article is written clearly.
The authors say what they want to do.

Experimental design

The datasets are collected from the DJI drone.
The article shows the ratio of training and testing datasets, 70 % of the sample data was used for the training model and 30% for evaluation.
The 3138 images have been used in this study.
It was collected at different times and dates
and this should be fine.

Validity of the findings

I think the findings are plausible and make sense.

Additional comments

I think the article is fine. However, if I may, I would like to suggest this to the authors.
1. Regarding deep learning performance well perform,
the article should give the reasons.
Why doesn't this study apply state-of-the-art deep learning models
as a classifier such as CNN or Mark-CNN?
2. Please provide an image size of this research.
3. The authors used SVM as a classifier but they don't show an objective function.
I suggest showing the equation and giving a few details of the objective equation of SVM in this research.
4. There is very little information about each classifier. Please provide more details about each classifier.
5. The CA symbol in the F-score equation shown in lines 249-251
is not described.
6. In line 179, the flowchart of this research is shown in figure 2. However, figure 2 shows area 1.
please check figure 2.

---

## Round 0.2 · accepted · Accept

The article has been improved after revisions and is accepted for publication.

Reviewer 1 ·

Basic reporting

Authors addressed all questions and queries.

Experimental design

It is now improved. Authors improved experimental design

Validity of the findings

I asked for justification of results and authors rightly addressed those.

Additional comments

Paper is in accepted state from my side. I do not have any further comment

Reviewer 3 ·

Basic reporting

The article is written clearly.

Experimental design

no comment

Validity of the findings

no comment

Additional comments

no comment